# The Association between Influenza Vaccination and the Risk of SARS-CoV-2 Infection, Severe Illness, and Death: A Systematic Review of the Literature

**DOI:** 10.3390/ijerph17217870

**Published:** 2020-10-27

**Authors:** Marco Del Riccio, Chiara Lorini, Guglielmo Bonaccorsi, John Paget, Saverio Caini

**Affiliations:** 1Postgraduate Medical School in Public Health, University of Florence, 50134 Florence, Italy; 2Department of Health Sciences, University of Florence, 50134 Florence, Italy; chiara.lorini@unifi.it (C.L.); guglielmo.bonaccorsi@unifi.it (G.B.); 3Netherlands Institute for Health Services Research (Nivel), 3513 Utrecht, The Netherlands; J.Paget@nivel.nl; 4Molecular and Lifestyle Epidemiology Branch, Institute for Cancer Research, Prevention and Clinical Network (ISPRO), 50139 Florence, Italy; s.caini@ispro.toscana.it

**Keywords:** SARS-CoV-2, influenza vaccine, systematic review, infection, severity, risk

## Abstract

We reviewed the association between seasonal influenza vaccination and the risk of SARS-CoV-2 infection or complicated illness or poor outcome (e.g., severe disease, need for hospitalization or ventilatory support, or death) among COVID-19 patients. None of the studies that were reviewed (*n* = 12) found a significant increase in the risk of infection or in the illness severity or lethality, and some reported significantly inverse associations. Our findings support measures aimed at raising influenza vaccination coverage in the coming months.

## 1. Introduction

There has been an important debate recently in the scientific community and the media about the relationship between influenza vaccination and COVID-19. Influenza and COVID-19 are respiratory viral illnesses that may be clinically indistinguishable and tend to be life-threatening in largely overlapping population subgroups (e.g., the elderly and people suffering from chronic health conditions). Moreover, because they are respiratory virus illnesses, their peak of activity may occur in the same period of the year (i.e., winter months in temperate countries). Based on the above considerations, most health professionals have advocated in favor of strengthening influenza vaccination programs, arguing that rising vaccine coverage could help improve COVID-19 patient management by allowing easier differential diagnosis and reducing the overload of healthcare systems, particularly intensive care units (ICUs) [1].

Public health decisions should be based as much as possible on the best available evidence regarding any benefits and drawbacks that the proposed intervention may be expected to entail, and because the influenza season is rapidly approaching in the Northern Hemisphere, a literature review on this topic is urgently needed. Here, we conducted a systematic review of the articles that examined whether influenza vaccination affects the risk of being infected with the SARS-CoV-2 virus, and the risk of complicated illness or poor outcome (e.g., severe disease, need for hospitalization or ventilatory support, or death) among COVID-19 patients.

## 2. Materials and Methods

The literature search was performed on 31 August 2020 by interrogating the MEDLINE, Embase and medRxiv databases for both peer-reviewed and non-peer-reviewed articles in any language (as long as an English abstract was available), using the following string: “SARS-CoV-2 OR COVID” AND “influenza OR flu” AND “vaccine *”. All identified articles were first independently screened by two researchers (MDR and SC) based on their title, and any considered potentially eligible for inclusion by either researcher was then obtained and read in full text. Any disagreement on the eligibility of a given article was resolved by consensus.

The literature search was then extended to the reference lists of all of the articles that were obtained in full copy (regardless of their final inclusion in the review). To be eligible for inclusion, an article had to be an original report based on individual-level data; studies relying on aggregated data (e.g., ecological studies reporting correlations [2,3]) were not retained because of their higher risk of bias. Letters and commentaries with no original data were also discarded.

Information and data related to the study design, size, and outcome, the participants’ mean/median age, the laboratory method used (if applicable), and the main results of the study were retrieved. Risk ratios (RRs), odds ratios (ORs), or hazard ratios (HRs) were reported whenever available, in addition to the information about any statistical adjustments that were made. The study quality was assessed using the Newcastle–Ottawa Scale (NOS), according to which the risk of bias in a given study is classified as low (overall score 7 to 9), moderate/high (4 to 6), or very high (0 to 3) [4].

## 3. Results

The literature search identified 1619 non-duplicate entries, of which 1461 were excluded based on their title. The remaining 158 articles were read in full copy, and an additional article was identified in their reference lists. Finally, twelve independent articles met all inclusion criteria and were retained (Figure 1).

Overall, the studies had good methodological quality, with the risk of bias judged to be low (overall score 7–9) in eight of twelve studies (Appendix A). The lack of statistical adjustments and the reliance on self-reported assessment of exposure (i.e., vaccination status) were the main issues for the four studies considered at moderate/high risk of bias.

Seven articles [7,8,9,10,11,12,13] focused on the association between influenza vaccination and the risk of SARS-CoV-2 infection (Table 1): these encompassed a total of 242,323 subjects, of which 56.6% were contributed by Pawlowski et al. [12] and 32.6% by Vila-Córcoles et al. [13]. Most studies were based on subjects from the general population, with the exception of the two smallest studies, which included 203 firefighters and paramedics from the USA [8], and 640 liver transplant patients from Italy [9]. The studies also differed in that they were based on a varying proportion of symptomatic and asymptomatic individuals. The laboratory diagnostic method varied across studies, with (qRT-) PCR being used in three of them. No statistical adjustment was made in four studies: Aziz et al. [7] and Donato et al. [9] found no significant association, whereas COVID-19 cases were significantly less likely to be vaccinated than test-negative subjects in the studies by Jehi et al. [10] (which separately reported on two independent cohorts) and Caban-Martinez et al. [8]. Three studies reported measures of association adjusted by age, gender, comorbidities, and other potential confounders, all of which found that influenza vaccinees were significantly less likely to become infected with the SARS-CoV-2 virus than non-vaccinees. Noale et al. [11] found a reduced risk among subjects aged less than 65 years (odds ratio (OR) 0.85, 95% confidence interval (CI) 0.74–0.98, *p* = 0.024; the OR among those aged ≥65 years was of similar magnitude, 0.87, but not statistically significant, *p* = 0.483). On the contrary, the association was stronger in the ≥65 years subgroup (relative risk 0.74, 95% CI 0.61–0.89, *p* < 0.01) in the study by Pawlowski et al. [12], and also achieved statistical significance (*p* < 0.03) in the whole population. Finally, a statistically significant inverse association (hazard ratio 0.63, 95% CI 0.43–0.92, *p* = 0.015) emerged among adults aged ≥50 years enrolled in the study by Vila-Córcoles et al. [13]. 

Five articles [6,14,15,16,17] reported on the association between influenza vaccination and the risk of severe illness and/or death among COVID-19 patients (Table 2). The total number of patients was 111,820, of which the majority (82.9%) were contributed by Fink et al. [6]. The latter was the only study encompassing a minority (16%) of non-laboratory-confirmed patients. Jehi et al. [14] and Murillo-Zamora et al. [15] found that the likelihood of being vaccinated against influenza was significantly (*p* < 0.001) or, respectively, nearly significantly (*p* = 0.073) lower among patients who required to be hospitalized compared to those who did not. Likewise, Fink et al. [6] reported a significantly lower odds of requiring intensive care or respiratory support among influenza vaccinees vs. non-vaccinees. The latter study also found that vaccinated COVID-19 patients were at significantly reduced risk of dying compared to non-vaccinated patients (OR 0.82, 95% CI 0.75–0.89, *p* < 0.01), but this finding was not confirmed in the studies by Ortiz-Prado et al. [16] and Poblador-Plou et al. [17].

## 4. Discussion

Influenza epidemics recur each year and the persistence of SARS-CoV-2 circulation in the upcoming months may expose healthcare systems to a severe risk of resource scarcity. Influenza vaccination is the cornerstone of influenza prevention, thus attaining higher vaccine coverage has been widely acknowledged as a public health priority [18]. Concerns about a possible association between influenza vaccination and the risk of coronavirus infection were raised based on Wolff’s paper, which examined endemic coronaviruses circulating in the USA in the 2017–2018 season, much earlier than the emergence of the SARS-CoV-2 virus [19]. After reviewing the existing literature on the topic, we can safely conclude that influenza vaccination is unlikely to be associated with an increase in SARS-CoV-2 risk of infection or with COVID-19 severity and the risk of associated death. In fact, most reviewed studies detected an inverse relationship, which was unexpected and even disconcerting given that influenza vaccines are not designed to protect from SARS-CoV-2. 

The studies included in this review are heterogeneous in many aspects, including their design, sample size, and inclusion criteria. In terms of study populations, some studies included individuals of all ages, while others focused only on the adult population. Unfortunately, only a few studies focused specifically on high-risk groups. In fact, only Noale et al. [11] and Pawlowski et al. [12] reported results specific to participants of ≥65 years, and no study reported on pediatric populations. In addition, the study by Donato et al. [9] was based on a population of transplanted patients, and caution is required in extrapolating its results to the general population. Moreover, the studies included in this review [7,8,9,10,11,12,13] used different testing criteria to assess the presence of SARS-CoV-2 infection, depending on the testing policy implemented in the area in which the study was conducted, and different diagnostic tests (of the seven studies specifically focusing on this aspect, three used molecular diagnosis with PCR [10,12,13], two performed antibody-based tests [7,8], and two did not specify the diagnostic tool that was used [9,11]). Furthermore, it must be acknowledged that all reviewed studies are retrospective and observational in nature, and thus likely to be subject to bias, and that not all studies reported measures of association adjusted by the relevant confounders. Finally, influenza vaccine effectiveness is difficult to estimate accurately and fluctuates across years, which may represent a further source of uncertainty in interpreting and comparing the results of the different studies [20]. Because of these several limitations, we recommend that further studies be conducted to confirm these preliminary findings and examine their validity in different population subgroups.

## 5. Conclusions

In conclusion, our review finds that, based on our knowledge (until the end of August 2020), public health measures aimed at raising influenza vaccine coverage should be encouraged. There is no evidence to suggest that this would have a negative impact on populations in terms of SARS-CoV-2 related infections, illness, or deaths.

## Figures and Tables

**Figure 1 ijerph-17-07870-f001:**
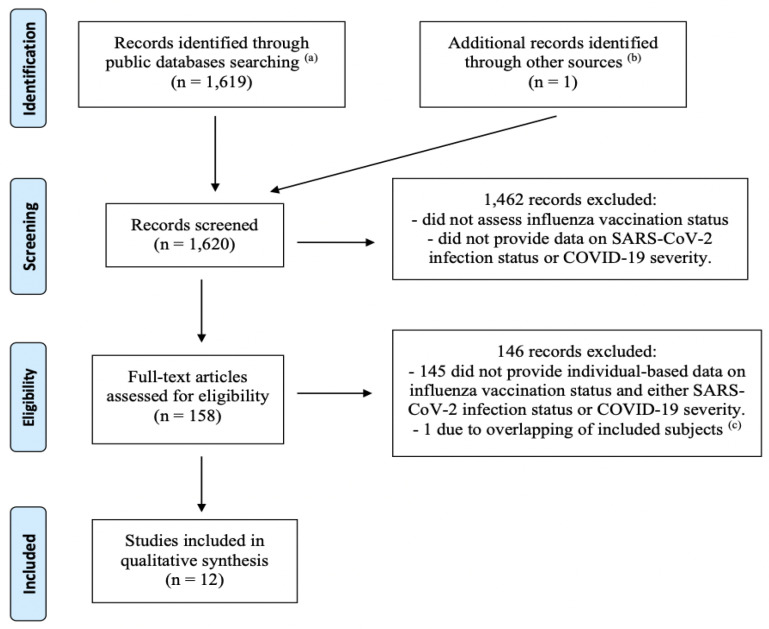
Flow-chart of the literature review (up to 31 August 2020) of studies investigating the association between influenza vaccination status and either the risk of being infected with the SARS-CoV-2 virus, or the risk of severe illness or death among COVID-19 patients. ^(a)^ MEDLINE, Embase, MedRxiv. ^(b)^ Reference list of records found through public databases. ^(c)^ The study by Santos et al. [5] was excluded due to probable overlap of the study population with the article by Fink et al. [6]. Both studies were based on data from hospitalizations for COVID-19 patients in Brazil registered in a national surveillance system. The study by Santos et al. was less recent and based on a smaller population, and examined fewer outcomes, compared to Fink et al., and, unlike the latter, performed no statistical adjustments.

**Table 1 ijerph-17-07870-t001:** Main features and results of studies on the association between influenza vaccination and the risk of infection with SARS-CoV-2.

Author	Study Sample	Age (Years)	Laboratory Method	Main Result	Adjustment
Aziz et al. [7]	Subjects participating in a community-based cohort study in Bonn, Germany (*n* = 4755)	≥30 (mean 55.2, SD 13.6)	ELISA and PRNT	No statistically significant association (no details available)	None
Caban-Martinez et al. [8]	Frontline firefighters and paramedics of a fire department in Florida, USA (*n* = 203) (a)	≥21	Point-of-care IgM-IgG LFIA	COVID-19 cases were significantly less likely to be vaccinated than controls (*p* = 0.027)	None
Donato et al. [9]	Liver transplant patients in Italy (*n* = 640)	≥20 (median 63)	Not specified	No statistically significant association (*p* = 0.238)	None
Jehi et al. [10]	Subjects tested in several clinics in Ohio, USA (*n* = 11,672) (b)	Any age COVID-19 negative: median 46.9, IQR 31.6–62.9 COVID-19 positive: median 54.2, IQR 38.8–65.9	RT-PCR	COVID-19 cases were significantly less likely to be vaccinated than controls (*p* < 0.001)	None
	Subjects tested in several clinics in Florida, USA (*n* = 2295) (b)	Any age COVID-19 negative: median 56.0, IQR 41.9–67.5 COVID–19 positive: median 52.6, IQR 36.7–63.1		COVID-19 cases were significantly less likely to be vaccinated than controls (*p* = 0.011)	
Noale et al. [11]	Subjects years participating in a web-based survey in Italy (*n* = 6650) (c)	≥18 (mean 48.0, SD 1 7.7)	Not specified	Vaccinated subjects <65 years old were significantly less likely to be infected (OR 0.85, 95% CI 0.74–0.98, *p* = 0.024). No statistically significant association among subjects ≥ 65 years (OR 0.87, 95% CI 0.59–1.28, *p* = 0.483)	Age, gender, education, comorbidities, other
Pawlowski et al. [12]	Subjects who received SARS-CoV-2 testing at Mayo Clinic, USA (*n* = 137,037)	Any age	PCR	Subjects vaccinated in the past year were significantly less likely to be infected (RR 0.85, 95% CI 0.75–0.96, *p* = 0.03). The association was stronger in the ≥ 65 years subgroup (RR 0.74, 95% CI 0.61–0.89, *p* < 0.01).	Propensity score matching (d) and multiple comparison
Vila-Córcoles et al. [13]	All subjects tested at primary healthcare center in Tarragona area, Spain (*n* = 79,071)	≥50	RT-PCR	Vaccinated subjects were significantly less likely to be infected (HR 0.63, 95% CI 0.43–0.92, *p* = 0.015)	Age, gender, vaccination history, comorbidities

(a) 185 patients had information on influenza vaccination available and were included in the analyses reported here. (b) Criteria for testing were: any of recent travel history to high-risk area, symptoms of respiratory illness (cough, fever, flu-like symptoms), physician discretion, or known contact with a COVID-19 case, during 12–17 March 2020; any of age >60 years or <26 months, comorbidities, immune therapy, known contact with a COVID-19 case from March 18 afterwards. (c) A total of 198,828 subjects participated (on a voluntary basis) in the web-based survey, of which 6650 reported SARS-CoV-2 nasopharyngeal swab testing and were included in the analyses. (d) Covariates considered in the propensity score matching: demographics (age, gender, race, ethnicity), county-level COVID-19 incidence and test-positive rate, comorbidities, pregnancy, and number of other vaccines. ELISA: enzyme-linked immunosorbent assay. LFIA: lateral flow immunoassay. PRNT: plaque reduction neutralization test. RT-PCR: reverse transcriptase-polymerase chain reaction. HR: hazard ratio. OR: odds ratio. RR: relative risk. CI: confidence interval. SD: standard deviation. IQR: inter-quartile range.

**Table 2 ijerph-17-07870-t002:** Main features and results of studies on the association between influenza vaccination and the risk of severe illness or death among COVID-19 patients.

Author	Study Sample	Age (Years)	Outcome	Main Result	Adjustment
Jehi et al. [14]	Laboratory-confirmed COVID-19 patients in Ohio and Florida, USA (*n* = 2852) (a)	Any age Not hospitalized: median 50.8, IQR 35.8–64.4 Hospitalized: median 64.4, IQR 54.8–76.6	Severe illness	Severe cases (requiring hospitalization) were significantly less likely to be vaccinated than non-severe cases (*p* < 0.001) (b)	None
	Laboratory-confirmed COVID-19 patients in Ohio and Florida, USA (*n* = 1684) (a)	Any age Not hospitalized: median 45.6, IQR 30.5–65.9 Hospitalized: median 64.9, IQR 52.5–76.8	Severe illness	Severe cases (requiring hospitalization) were significantly less likely to be vaccinated than non-severe cases (*p* < 0.001) (b)	None
Murillo-Zamora et al. [15]	Laboratory-confirmed COVID-19 patients in Mexico (*n* = 740)	≥15 (mean 43.7, SD 14.9)	Severe illness	Severe cases (dyspnoea requiring hospitalization) were non-significantly less likely to be vaccinated than non-severe cases (*p* = 0.073)	None
Fink et al. [6]	All clinically confirmed COVID-19 patients in Brazil (*n* = 92,664, of which 84% laboratory-confirmed) (c)	Any age (median 59)	Severe illness	Vaccinated patients were significantly less likely to require intensive care (OR 0.92, 95% CI 0.86–0.99, *p* < 0.05) or respiratory support (OR 0.81, 95% CI 0.74–0.88, *p* < 0.01)	Age, SES, comorbidities, other
			Death	Vaccinated patients were at significantly reduced risk of death (OR 0.82, 95% CI 0.75–0.89, *p* < 0.01) (d)	
Ortiz-Prado et al. [16]	All laboratory-confirmed COVID-19 patients in Ecuador (*n* = 9468)	Any age Men: median 42, IQR 32–56 Women: median 39, IQR 30–54	Death	No statistically significant association (OR among vaccinated patients: 0.71, 95% CI 0.23–2.17)	Age, gender, comorbidities
Poblador-Plou et al. [17]	All laboratory-confirmed COVID-19 patients in Aragon, Spain, with follow-up ≥ 30 days (*n* = 4412)	Any age (mean 67.7, SD 20.7)	Death	No significant differences in the proportion of vaccination between deceased and alive patients after adjusting by age (*p* = 0.110 among men, *p* = 0.126 among women)	Age

(a) COVID-19 patients diagnosed from 8 March to 1 May were included in a “development” cohort (used to build a predictive model), and patients diagnosed afterwards (until 5 June) were included in a “validation” cohort (for model validation). Results were only provided separately for the two cohorts. (b) Influenza vaccination was included in the predictive model for the risk of hospitalization for COVID-19 patients. (c) The number of patients with available information on both vaccination status and the outcome of interest and that, therefore, were included in the analyses reported here were: 26,260 for the risk of requiring intensive care; 25,959 for the risk of requiring respiratory support; and 19,274 for the risk of death. (d) Analysis restricted to laboratory-confirmed COVID-19 cases only. Results were confirmed also when including non-laboratory confirmed patients (OR 0.84, 95% CI 0.77–0.91, *p* < 0.01). OR: odds ratio. CI: confidence interval. SES: socio-economic status. SD: standard deviation. IQR: inter-quartile range.

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
