# Peer review of "The Association between Influenza Vaccination and the Risk of SARS-CoV-2 Infection, Severe Illness, and Death: A Systematic Review of the Literature"

_ijerph, 2020, doi:10.3390/ijerph17217870_

Round 1

Reviewer 1 Report

This manuscript is a review article that examines the impact of influenza vaccination upon the likelihood of infection with SARS-CoV-2 and/or development of serious COVID-19 disease.  The authors sifted through a large body of literature using relevant keywords, then they settled upon 12 research articles that looked for correlations between the aforementioned circumstances.  

My comments are mostly minor in nature, because this is a fairly short, straightforward manuscript.  

1.  The title to Table 1 suggests that the authors were examining an association between flu vaccination and "the risk of infection with SARS-CoV-2".  However, in the "Main Result" column it seems that the main endpoint was not actually infection with SARS-CoV-2, but the development of any diseases symptoms of COVID-19.  The main difference between these two endpoints would be asymptomatic individuals; these people would add to the infected category but would be invisible to the COVID-19 disease group.  Some studies have been performed to look for evidence of SARS-CoV-2 infection in populations not experiencing symptoms, which in my mind would be necessary to report results under the Table 1 title.  My recommendation would be to change the name of Table 1 to make it more accurate to COVID-19 disease of any sort, unless I am misunderstanding the information presented in the table.  

Grammatical errors:  

  1. Line 24-25: delete the text "and in the media"
  2. The sentence beginning on line 49 and ending on line 52 appears to be incomplete.  "because of their higher proneness to bias and confounding......"  Confounding what?  
  3. The acronym on line 63 is not written in a conventional way.  qRT-PCR seems to be the more common acronym.  Solely using RT is confusing because it can refer to Reverse Transcriptase or Real Time.  
  4. On line 116, the words "resources scarcity" should be changed to "resource scarcity"
  5. On line 119, the text reading "based on the Wolff's paper" should be changed to "based on Wolff's paper".  

Reviewer 2 Report

This is a nice short communication for a systemic review of the literature for the association between influenza vaccination and the risk of SARS-CoV-2 infection, severe illness, and death.   I have the following comments/suggestions.

  1. Line 24-25: "in the media" was used twice in the same sentence.  Consider revision.
  2. In Materials and Methods: please indicate what did you do with articles with different languages.
  3. Please include age in all studies shown in Table 1 and 2
  4. Do you have studies that included patients <18 years?
  5. Do you have studies that focused on the high risk age group eg <2-5 years or >65 years
  6. Conclusions must be more specific to the information you reviewed.   If this can be applied to all age group or adult only.  If this review included mostly healthy individual.    I saw one study that you included in the review with liver transplant and no significant association (while other studies showed significant association).  Do we have enough information to say about patients with other risk factors such as obesity, asthma, diabetes, etc. 

Reviewer 3 Report

See archive

Round 2

Reviewer 2 Report

Thank you for the revised version of the manuscript.  I have no further comments.

Reviewer 3 Report

No comments